# Dispatch Optimization Model for Haulage Equipment between Stopes Based on Mine Short-Term Resource Planning

**Ning Li** [1,2,*] [iD]**, Shuzhao Feng** [1,*]**, Haiwang Ye** [1,2]**, Qizhou Wang** [1,2,*]**, Mingtao Jia** [3]**, Liguan Wang** [3]**, Shugang Zhao** [4] **and Dongfang Chen** [1,2]

1   School of Resource and Environment Engineering, Wuhan University of Technology, Wuhan 430070, China; yehaiwang@sina.com (H.Y.); chendongfang@whut.edu.cn (D.C.)
2   Hubei Key Laboratory of Mineral Resources Processing and Environment, Wuhan 430070, China
3   School of Resource and Safety Engineering, Central South University, Changsha 410083, China; mingtao_jia@163.com (M.J.); 13808478410@163.com (L.W.)
4   MCC Zhicheng (Wuhan) Engineering Technology Co., Ltd., Wuhan 430074, China; zhaoshugangkaka@outlook.com
*   Correspondence: 13875910191@163.com (N.L.); fsz2224284697@163.com (S.F.); wqz@whut.edu.cn (Q.W.); Tel.: +86-13875910191 (N.L.)

**Abstract:** The working environment of underground mines is complicated, making it difficult to construct an underground mine production plan. In response to the requirements for the preparation of a short-term production plan for underground mines, an optimization model for short-term resource planning was constructed, with the goal of maximizing the total revenue during the planning period. The artificial bee colony optimization algorithm is used to solve the model using MATLAB. According to the basic requirements of underground mine ore haulage and ore hoisting, a haulage equipment inter-stopes dispatch plan model was constructed, with the primary goal of minimizing the haulage equipment wait time. A non-dominated sorting genetic algorithm is used to solve the optimization model. An underground mine is examined using the two models, and the optimization results are compared and verified with the scheme obtained by using traditional optimization algorithms. Results show that based on the improved optimization algorithm, the use of short-term production planning schemes to guide mine production operations can increase the haulage equipment utilization rate, thereby increasing mine production revenue.

**Keywords:** digital mine; mine short-term production planning; haulage equipment dispatch plan; ABCA; NSGA

## 1. Introduction

The complete mining cycle of a mine can be divided into two categories: the development period and production period. During development, a series of tunnels from the surface to the ore body is excavated to establish a passage between the surface and ore body for transporting personnel, equipment, materials, ore and waste rock, as well as creating appropriate conditions for mining and forming an independent system for hoisting, transportation, ventilation, drainage, power supply, and water supply, within these tunnels. The main material to be transported is generally rock, which has a low economic value. Investments during this period are primarily used to purchase equipment and hire manpower; therefore, the mine operates at a negative profit during the development period. After mine development and any necessary planning are completed, the mine enters the production period. The haulage is primarily composed of ore with high economic value. The production period is divided into three periods: ramp-up, steady state and tailing-off periods. Early revenue is to offset the costs from the development period, and mines begin to make profit during the middle-term; however, in the last term, a portion of the profit must be used to maintain the stability of the ore body. In order to maximize profit during

the production period, Topal [1] and Sandanayake et al. [2] considered mining process allocation, comprehensive scheduling, stope design, and construction of system planning.

Different mining cycles correspond to different mining plans. Wu and Li [3] pointed out that mine production planning can be divided into long-term, medium-term, and short-term planning according to the length of the planning time. The length of the time period for each planning type is different, leading to different optimization goals. Long-term planning defines the company's long-term goals, emphasizing timeliness and strategy, and fully considers market changes; medium-term planning is guided by the long-term goals to complete the annual plan; and short-term planning can be divided into four plan types according to the length of time: seasonal planning, monthly planning, weekly planning, and daily planning, including product output, operation arrangement, labor allocation, grade control, product sales, and haulage cost. Planning the mine location and scheduling the mining operations is conducted during the production period. In short-term planning, resource planning is aligned to the mining cycle, and daily planning is generally aimed at equipment dispatch planning.

Previous scholars' research on short-term production in underground mines only divided the mining areas of the mine at each period, and did not involve the actual mining process. The production dispatching planning of underground mining vehicles is a short-term production planning process that is different from short-term resource planning. Only after short-term resource planning is completed, can production dispatching planning begin. However, most short-term production planning does not involve precise equipment dispatch path optimization research. This article combines resource planning and dispatching path optimization to guide the actual short-term production of a mine.

Newman et al. [4] pointed out that an underground mine contains more influencing factors than an open-pit mine, and discussed the advantages of mine production optimization. Based on their work, this article presents two mathematical models for optimizing different types of short-term production planning in underground metal mines and presents different optimization algorithms to obtain better short-term production planning schemes.

In the following sections, the data set used to test the model calculation results is described. Next, the mathematical models used in the construction of different production plans and their corresponding solving algorithms are introduced. The results are compared to the operational plan obtained from the traditional optimization algorithm.

### 1.1. Terms and Concepts

Every industry has its own unique terminology to describe its operating status and industry characteristics, and the mining industry is no exception. This section briefly introduces technical terms and concepts mentioned in this article.

In short-term production planning, weekly planning generally refers to resource arrangement, and daily planning generally refers to dispatching planning of production equipment. The excavation can start only after vehicle dispatching planning is completed.

During resource planning, the overall profit of the mine is maximized as much as possible while satisfying the constraints and medium-term planning objectives. The purpose of constructing a haulage equipment dispatch plan is to complete the resource planning scheme more efficiently. This process does not consider the benefits and objectives. Making the equipment utilization rate as high as possible while meeting the haulage conditions is necessary. Short-term production planning combines resource planning with equipment dispatch planning and can plan daily production tasks more accurately as well as enable the mine to perform production operations efficiently.

### 1.2. Dataset

In order to test the validity of the mathematical model and optimization algorithm, actual data from an iron mine (Figure 1) in Handan City, Hebei Province, People's Republic of China is analyzed. The ore body of the mine is approximately 1400 m long and 300 m

wide. Here, we will briefly introduce some mining work and equipment involved in short-term production operations of underground mines.

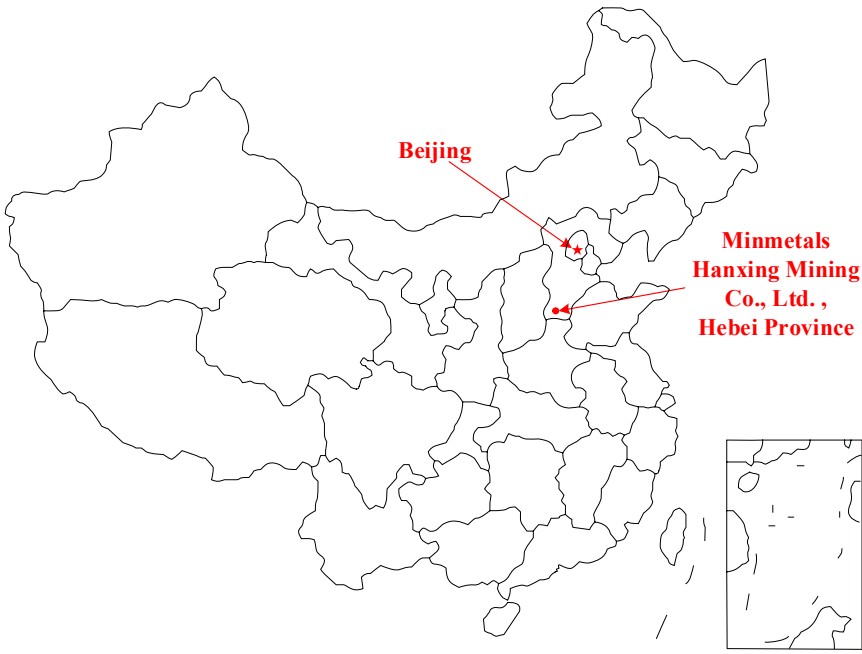

**Figure 1.** Location map.

1.2.1. Mine Layout

After the mine ground construction and road repairs are completed, ore extraction begins with the excavation of the main shaft. When the depth of the main shaft reaches the depth of the ore body, a shaft station connecting the transport roadway and the beginning of the main shaft is excavated. Generally, the shaft station chambers have storage lines, traffic lines, shunting lines, water pump chambers, substations, dispatching rooms, and repair warehouses. The transportation tunnel is placed between the ore body and the shaft station and is used for transporting ore and ventilation. Cross-cuts are excavated horizontally as links between the levels and the main shaft. The transportation tunnels that are parallel to the run of the ore vein are called drifts, and those vertical to the run of the ore vein are called crosscuts (Figure 2).

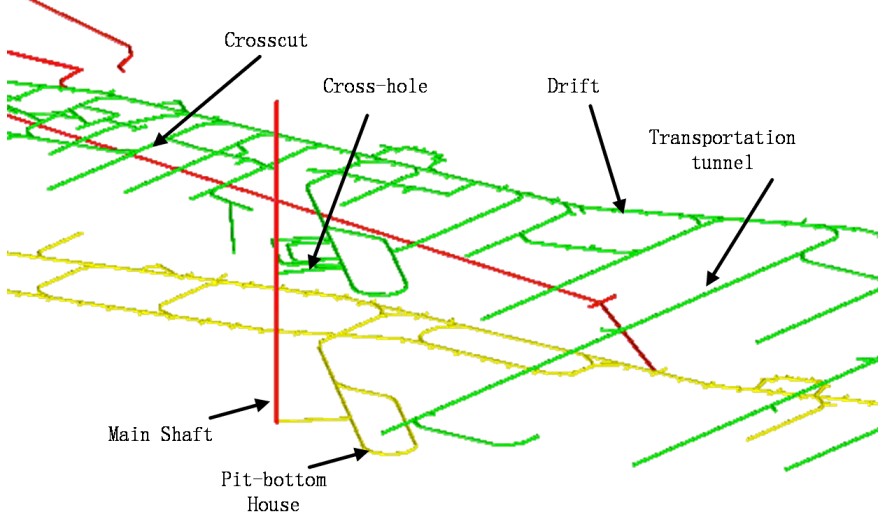

**Figure 2.** Mine layout.

### 1.2.2. Mining Methods

All or part of the mineral deposits mined by a mining enterprise is called mine fields. Within the mine fields, the main haulage tunnels that are consistent with the strike of the ore body are excavated within certain vertical distance, and the mine fields are divided vertically into several mine sections. These mine sections are called levels, which are divided into a number of independent mining units according to a certain size, called ore blocks.

The selection of mining methods for underground mines should consider multiple factors, such as location, thickness, shape, occurrence, and geological conditions of the ore body. Different mining methods have significant differences in mining rules, ore output, and production capacity of mining equipment. Based on the above factors, the mine chose to adopt the cut-and-fill mining method.

The cut-and-fill mining method commonly divides an ore block into an ore pillar and chamber, and the mining order is the chamber first, then the ore pillar. The chamber is excavated from bottom to top by sublevels, then filled sequentially to maintain the surrounding rock of the upper and lower walls to create continuous upward mining operation conditions. Top filling is carried out when the chamber reaches the last sublevel. After several chambers or whole levels have been mined, the ore pillar begins to be mined. This method is a working face cycle operation. After one cycle of rock drilling and blasting, ore extraction, backfilling, and roof protection is completed, the next sublevel cycle is carried out.

During the medium-term planning of the mine, mining two levels at the same time are necessary, each level has two sublevels, and each sublevel has 36 sites, 12 along the X-axis direction and three along the Y-axis direction. Short-term resource planning needs to determine which sites within the two levels can be used as stopes for production operations, and haulage equipment dispatch planning needs to reasonably arrange vehicles for haulage operations in different levels. The short-term planning period is seven days. During the planning period, the output ore volume is approximately 40,000 tons, the market metal price of iron is 488 dollars/t, the mining cost is 38.6 dollars/t, the ore recovery ratio is 91%, and the mineral processing recovery ratio is 81%.

### 1.2.3. Spatial Relations

There is no restriction on the dimensions of each mine point in an open-pit mine. Due to the special mining environment of underground mines, the selection of sites needs to be determined based on the mining method as well as the temporal and spatial development of the ore block.

(1) During the same planning period, multiple levels are mined simultaneously. Each level is divided into multiple ore blocks and subsequently divided into chambers and ore pillars. The chambers are stoped in sublevels from bottom to top. Ore blocks in each sublevel have unique attributes such as grade and total volume of ore and waste. (2) During the early mining stage, in order to make the mine recuperate costs as soon as possible, the ratio of ore mining in the upper levels to the lower levels should be greater than one. (3) In order to ensure safe mining, during the same mining period, the ore blocks at the same horizontal position in each level are stoped within a given sublevel from bottom to top. The mining conditions are classified into three types: (1) The lower sublevel ore block is not mined, and mining of the upper sublevel ore block is prohibited; (2) the lower sublevel ore block is being mined, and mining of the upper sublevel ore block is prohibited; and (3) the lower sublevel ore block has been mined, and the upper sublevel ore block can be mined. (4) As shown in Figure 3, in the horizontal direction within each sublevel, only one of the four adjacent ore blocks can be mined in the same mining period. This is to avoid excessively large exposed areas of the roof between sublevels, which may cause stress concentrations and accidents.

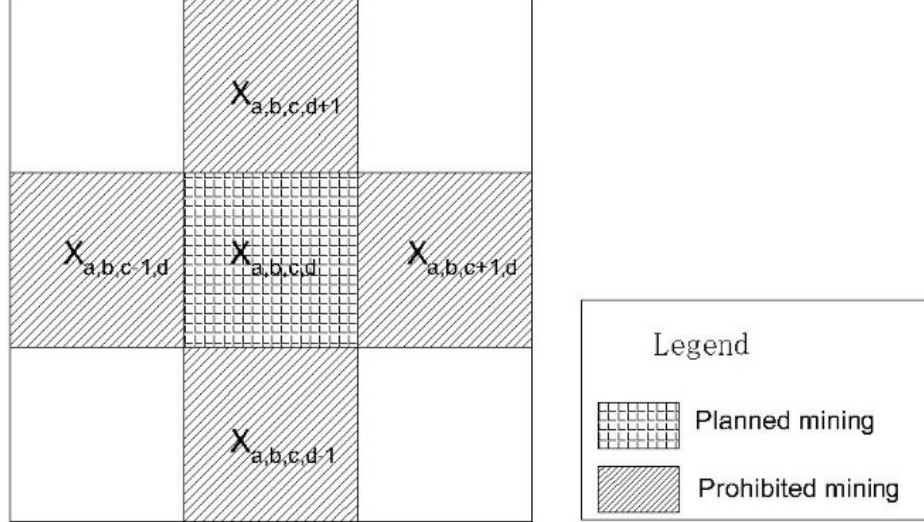

**Figure 3.** Horizontal direction mining rule.

### 1.2.4. Haulage Equipment

Currently, the production equipment of underground mines is gradually becoming trackless. Zhan [5] analyzed the current research and application status of trackless mining equipment as well as comprehensively reviewed the main problems of trackless mining equipment in the People's Republic of China and the development trend of trackless mining equipment technology. The primary equipment for underground mining operations includes rock drills, scrapers, electric locomotives, and bogies. With the exception of rock drills, the main purpose of the other equipment is to transport ore. Yi [6] pointed out that the scraper has become a core piece of trackless equipment. The scraper is responsible for transporting ore from the stope to the ore pass on each sublevel. An electric locomotive is used to transport the ore from the lowest sublevel of each level to the main shaft for hoisting. In the mine mentioned in this article, six scrapers with the same specifications and two electric locomotives with the same specifications are used for production operations during each planning period. The load capacity of the scraper is 3 t, and each electric locomotive can pull 13 mine bogies with a load capacity of 4 t each.

Compared to rock drills, scrapers and electric locomotives are more flexible to use. They consistently engage in irregular movements, making the operation time and route difficult to predict. A reasonable dispatching plan for haulage equipment is required. In addition, the operational planning of scraper and the electric locomotive is affected by a variety of constraints. The most basic constraint is the waiting time of the scraper, followed by the mining volume requirements of the mining and unloading sites, as well as the scraper's running route and speed limit within the complex roadway. Under these constraints, the scraper and electric locomotive are not restricted to only move back and forth between the fixed mining and unloading sites, allowing for a variety of dispatching options in each day. The purpose of this article to optimize the dispatching planning of haulage equipment by selecting a scheme that can minimize the total equipment waiting time, improve the equipment utilization of haulage equipment, and further reduce production costs. The simple dispatching model of haulage equipment is shown in Figure 4.

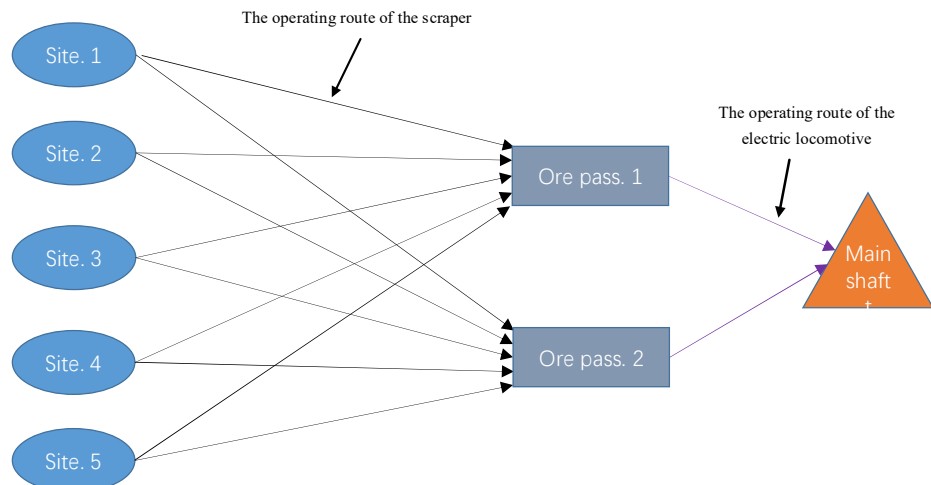

**Figure 4.** Simple dispatching model.

## 2. Relevant Literature

### 2.1. Short-Term Resource Planning

For underground mines, long-term and medium-term planning goals are set by the enterprise or decision-makers, with a wide range of goals and relatively few constraints. However, short-term planning is closely related to actual mine operations, which requires precise targets and tight plans. It is difficult to realize short-term planning by relying solely on decision support systems based on experience. Therefore, a mathematical model is needed to solve the short-term production planning problem by considering it as two constrained optimization problems that are long-term and medium-term planning goals, as well as constraints of the mine itself. The mathematical model is solved using optimization algorithms to quickly and effectively obtain a better short-term production scheme.

Li and Tong [7] proposed that the main evaluation methods for underground mine planning are optimization and simulation. The simulation method has strong applicability, while the optimization method has higher theoretical and application value. In the optimization method, scholars visualize the resource planning problem as a corresponding mathematical model or ore allocation problem, accounting for the various constraints using linear programming, multi-objective programming, and dynamic programming in Operational Research methods to determine the solution. Ming et al. [8] established a mineral product market demand planning model based on the production characteristics and operation of underground metal mines. In addition to the methods mentioned above, Nehring et al. [9] adopted a single mathematical optimization model combining the short-term goal of minimizing the deviation of the feed grade of the target concentrator with the medium-term goal of maximizing the net present value, to solve the short-term and medium-term integrated production planning tasks. O'Sullivan and Newman [10] established a mathematical model with both resource constraints and priority constraints, and then used a heuristic method to reduce the model specifications. They also constructed an optimization-based decomposition heuristic method to solve the model.

Based on operational research methods, in order to speed up the solution process, intelligence algorithms have gradually become the mainstream solution for solving mathematical models. Zhou and Gu [11] proposed the application of genetic algorithms to optimize the mining sequence for underground mines based on numerical simulations of the mining sequence. Hou et al. [12] considered the technical and economic requirements and the spatial sequence relationship in the mining process, constructed a dynamic optimization model for the production planning of polymetallic underground mines with the goal of maximizing profit, and gave a solution algorithm based on the artificial bee colony model. Foroughi et al. [13] constructed a multi-objective integer programming model that introduced a non-dominated sorting genetic algorithm to solve the target model. The

algorithm showed good convergence and diversity, and the solution time was significantly reduced. Gligoric et al. [14] expressed the ore body as a set of minable blocks based on the establishment of a production planning model, and applied a multi-objective iterative greedy algorithm to define a set of minable blocks each year to make the deviation from the target less than or equal to the given minimum error.

With the continuous development of computer hardware, three-dimensional visualization technology has been gradually applied to resource planning. Jiang et al. [15] proposed a construction method for three-dimensional visualization production plan based on simulated mining technology by using DEMINE. In order to resolve the issue that operations research methods or optimization algorithms cannot be connected with 3D visualization technology, Liu et al. [16] used multi-objective planning, combined with logical constraints, business constraints, and spatial constraints, as well as established a multi-objective planning model using three-dimensional visualization technology to show the spatial logical relationship of mine engineering.

Other methods can be also used to solve production planning problems. Sarin and West-Hansen [17] proposed a model to optimize the start-up time of different parts of underground coal mines. Newman and Kuchta [18] constructed a mixed integer program to plan ore production over multiple time periods. Riff et al. [19] constructed a "greedy random adaptive search" program to speed up the model solving process for copper mines. Little et al. [20] showed the value of optimizing the shape of the stope by providing a model to optimize the shape of the two stopes. Mousavi and Sellers [21] integrated in mine recovery (IMR) into conventional mining operations, which can significantly increase the net present value of the project by recycling low-grade material from conventional mining that is commonly left as waste rock. Campeau and Gamache [22] proposed an optimization model for short-term planning, taking into account the various working points from the development and production phases, as well as the specific equipment and worker restrictions using a mixed integer program with priority. Gligoric et al. [23] proposed a long-term mine planning method for underground lead-zinc mines based on fuzzy logic aimed at the production plan of lead-zinc mines under uncertain conditions. A fuzzy stochastic inventory control model was established.

### 2.2. Haulage Equipment Dispatch Planning

Compared with research on the short-term production planning of underground mines, there are relatively few studies on the production dispatching planning of underground mine vehicles. Gamache et al. [24] proposed a solution based on the shortest-path algorithm, in which each decision of the solution accounts for the current state of the mine. With automation comes the possibility for optimization. Saayman [25] looked at possible solutions to the problem of optimizing the autonomous vehicle dispatch system in an underground mine, and evaluated possible optimization strategies using a simulated environment. Nehring et al. [26] proposed a classic mixed integer programming model to optimize the production dispatching of the sublevel shutdown process and proposed a new model formula that can significantly reduce the solution time and maintain all constraints without changing the results.

The optimization problem of the vehicle path can be regarded as a non-polynomial problem. The optimal solution cannot be derived directly. It can only be verified by polynomials whether the proposed solution is a feasible solution, and then find the optimal solution. Sun and Lian [27] used the ant colony algorithm to solve this NP problem to optimize the vehicle haulage route of a certain shift in underground mine production dispatching, and this algorithm first used adaptive strategy to control its convergence speed, which can improve search performance and then optimize performance indicators based on the traditional ant colony algorithm during each iteration. Åstrand et al. [28] proposed a constraint programming method that can automatically realize the short-term dispatching process of cut-and-fill mines. This method builds on previous work by considering the running time of the fleet.

## 3. Model

Since the short-term production plan of the underground mine mentioned in this article includes a short-term resource plan and a dispatching plan for haulage equipment, two different models are needed to construct the different plans. The following will focus on the selection of objective functions and constraints of the two models. According to the time span used by most mines and planners, the short-term resource plan cycle in this paper is one week, and the haulage equipment dispatch plan cycle is 8 h in a shift.

### 3.1. Model of Short-Term Resource Planning

The underground mine short-term resource plan is based on the medium-term and long-term plans. First, determine the time span of the short-term plan, then determine the planned mining volume within the time span of the short-term plan, and finally determine the short-term mining scheme based on the medium-term and long-term plan. The purpose of a short-term resource plan is to determine the mining sequence in most levels. The optimization model established by the traditional 0-1 integer programming method can solve this problem.

#### 3.1.1. Sets

$\mathcal{A}$: Set of levels mined simultaneously during the planning period.

$$\mathcal{A} = \{1, \ldots, A\}, a \in \mathcal{A}$$

$\mathcal{B}$: Set of sublevels divided during the mining of the whole block.

$$\mathcal{B} = \{1, \ldots, B\}, b \in \mathcal{B}$$

$\mathcal{C}$: Set of ore blocks in the x direction on each sublevel.

$$\mathcal{C} = \{1, \ldots, C\}, c \in \mathcal{C}$$

$\mathcal{D}$: Set of ore blocks in the y direction on each sublevel.

$$\mathcal{D} = \{1, \ldots, D\}, d \in \mathcal{D}$$

#### 3.1.2. Parameters

$P$: Market price of iron;
$W_{a,b,c,d}$: Total tonnage of ore and waste in the (a, b, c, d) ore block;
$G_{a,b,c,d}$: Metal grade in the (a, b, c, d) ore block;
$R$: Ore recovery ratio;
$\varepsilon$: Dressing recovery ratio of the metal;
$C_{a,b,c,d}$: Mining cost of the (a, b, c, d) ore block;
$maxG$: The upper limit of the grade required by the dressing plant of the metal mining enterprise;
$minG$: The lower limit of the grade required by the dressing plant of the metal mining enterprise;
$Q_{max}$: The planned maximum tonnage of ore and waste during the plan period;
$Q_{min}$: The planned minimum tonnage of ore and waste during the plan period;
$N_{a,b}$: The number of scrapers and rock drills in b sublevel, a level.

#### 3.1.3. Variables

$X_{a,b,c,d}$ is a binary variable, indicating whether the (a, b, c, d) ore block will be mined in the short-term plan.

$$X_{a,b,c,d} = \begin{cases} 1, & \text{The ore blocks are mined during the plan period} \\ 0, & \text{The ore blocks will not be mined during the plan period} \end{cases}$$

### 3.1.4. Objective

When constructing the short-term resource plan for the mine, the layout of the block should be rationally planned according to the current metal market price, combined with the grade of each mining ore block. Then, the difference between the income and cost of all mining blocks during the planning period is taken as an objective function to maximize the total production income of the mine in each short-term plan period. The first part of Equation (1) is the sum of the income of all mining blocks during this planning period. The second part of the equation is the sum of the mining costs during the plan period.

$$maxP = \sum_{a \in \mathcal{A}} \sum_{b \in \mathcal{B}} \sum_{c \in \mathcal{C}} \sum_{d \in \mathcal{D}} P W_{a,b,c,d} G_{a,b,c,d} R \varepsilon X_{a,b,c,d} - \sum_{a \in \mathcal{A}} \sum_{b \in \mathcal{B}} \sum_{c \in \mathcal{C}} \sum_{d \in \mathcal{D}} C_{a,b,c,d} X_{a,b,c,d} \quad (1)$$

Under the premise of meeting the mine production requirements, the mine can obtain greater profits during each plan period. This is the goal pursued by every planner, to maximize the economic benefits of the mine and provide funds for later production or purchase advanced equipment, thereby improving the efficiency and economic benefits of subsequent production.

### 3.1.5. Constraints

Constraint (2) determines the tonnage of ore produced during the plan period so that it cannot be greater or less than a predetermined tonnage, which would affect the production process of the mine's medium- and long-term plan. Constraint (3) is the restriction on the dressing grade. In order to meet the grade requirements for the dressing plant, the grade of the mined ore should be guaranteed within a certain range; otherwise, the dressing recovery rate will decrease. Due to the current way of work, constraint (4) determines the number of ore blocks mined in each sublevel during the plan period, which should be greater than the total number of scrapers and rock drills in that sublevel. This constraint can effectively prevent production equipment from waiting in the stope and improve equipment time utilization ratio. Constraint (5) restricts the mined total tonnage of ore and waste between the levels, making the mined total tonnage of ore and waste in the upper level in the early stage greater than that in the lower level. Next, Constraints (6) and (7) are spatial relationship constraints in the vertical direction. Constraint (6) means that during the same plan period, the lower sublevel ore blocks are not mined, and the ore blocks with the same horizontal position in the upper sublevel are not mined. Once mining of the lower sublevel ore blocks is completed, the ore blocks with the same horizontal position of the upper sublevel can be mined. Constraint (7) restricts simultaneous mining of ore blocks in adjacent sublevels with the same horizontal position. Constraint (8) is a horizontal spatial constraint that specifically states that in the same plan period, only one of the four adjacent ore blocks in the same sublevel can be mined.

$$Q_{min} \leq \sum_{a \in \mathcal{A}} \sum_{b \in \mathcal{B}} \sum_{c \in \mathcal{C}} \sum_{d \in \mathcal{D}} W_{a,b,c,d} X_{a,b,c,d} \leq Q_{max} \quad (2)$$

$$minG \leq \frac{\sum_{a \in \mathcal{A}} \sum_{b \in \mathcal{B}} \sum_{c \in \mathcal{C}} \sum_{d \in \mathcal{D}} W_{a,b,c,d} G_{a,b,c,d} X_{a,b,c,d} R}{\sum_{a \in \mathcal{A}} \sum_{b \in \mathcal{B}} \sum_{c \in \mathcal{C}} \sum_{d \in \mathcal{D}} W_{a,b,c,d} X_{a,b,c,d}} \leq maxG \quad (3)$$

$$\sum_{c \in \mathcal{C}} \sum_{d \in \mathcal{D}} X_{a,b,c,d} \geq N_{a,b} \quad (4)$$

$$\sum_{b \in \mathcal{B}} \sum_{c \in \mathcal{C}} \sum_{d \in \mathcal{D}} W_{a,b,c,d} X_{a,b,c,d} \geq \sum_{b \in \mathcal{B}} \sum_{c \in \mathcal{C}} \sum_{d \in \mathcal{D}} W_{a+1,b,c,d} X_{a+1,b,c,d} \quad (5)$$

$$X_{a,b,c,d} - X_{a,b+1,c,d} \geq 0, a \in [1, A], b \in [1, B-1], c \in [1, C], d \in [1, D] \quad (6)$$

$$X_{a,b,c,d} + X_{a,b+1,c,d} \leq 1, a \in [1, A], b \in [1, B-1], c \in [1, C], d \in [1, D] \quad (7)$$

$$X_{a,b,c,d} + X_{a,b,c-1,d} + X_{a,b,c+1,d} + X_{a,b,c,d-1} + X_{a,b,c,d+1} \leq 1, a \in [1, A], b \in [1, B], c \in [2, C - 1], d \in [2, D - 1] \quad (8)$$

*3.2. Haulage Equipment Dispatch Plan Model*

Haulage equipment in the mine under study includes scrapers and electric locomotives. The scrapers are responsible for hauling the ore from the stope of each sublevel to the ore pass. The electric locomotive hauls the ore from the ore pass of each level to the main shaft for hoisting. After the short-term resource plan is created, the scrapers will alternately operate among the sites. It is not possible to make a scraper only run back and forth between specific stopes and ore passes, which will affect the production balance and efficiency of the short-term plan.

The most important parameter in the dispatch plan model is time, including the loaded and empty travel time of the scraper as well as the loaded and unloaded travel time of the electric locomotive. Only by determining each operating time can a haulage equipment dispatch plan be constructed.

3.2.1. Sets

$\mathcal{A}$: Set of levels mined simultaneously during the plan period.

$$\mathcal{A} = \{1, \ldots, A\}, a \in \mathcal{A}$$

$\mathcal{B}$: Set of sublevels divided during the mining of the whole block.

$$\mathcal{B} = \{1, \ldots, B\}, b \in \mathcal{B}$$

$\mathcal{C}_{ab}$: Set of scrapers in b sublevel, a level.

$$\mathcal{C}_{ab} = \{1, \ldots, C_{ab}\}, c \in \mathcal{C}_{ab}$$

$\mathcal{D}_a$: Set of electric locomotives in a level.

$$\mathcal{D}_a = \{1, \ldots, D_a\}, d \in \mathcal{D}_a$$

$\mathcal{N}_{ab}$: Set of stopes in b sublevel, a level.

$$\mathcal{N}_{ab} = \{1, \ldots, N_{ab}\}, i \in \mathcal{N}_{ab}$$

$\mathcal{M}_a$: Set of ore passes in a level.

$$\mathcal{M}_a = \{1, \ldots, M_a\}, j \in \mathcal{M}_a$$

3.2.2. Parameters

$T_{limit}$: Working hours in a shift, generally 8 h;

$z_{abcij}$: The loaded travel time of c scraper from i stope to j ore pass in b sublevel, a level;

$q_{abcij}$: The empty travel time of c scraper from i stope to j ore pass in b sublevel, a level;

$z_{adj}$: The loaded travel time of d electric locomotive from j ore pass to main shaft in a level;

$q_{adj}$: The empty travel time of d electric locomotive from j ore pass to main shaft in a level;

$R_{abc}$: The capacity of c scraper in b sublevel, a level;

$g_{abj}$: The total tonnage of ore and waste of i stope in b sublevel, a level;

$h_{abj}$: The maximum number of loading vehicles in one shift at i stope in b sublevel, a level;

$J_{aj}$: The maximum ore draw of j ore pass in a level, that is, the tonnage of ore drawn when the ore has been drawn in one shift;

$R_{ad}$: The capacity of d electric locomotive in a level;

$F_{aj}$: The drawing capacity of j ore pass in a level, that is, the general tonnage of ore drawn obtained from a statistical analysis of the general tonnage of ore drawn from multiple ore passes in a shift;

$Q_{ajmax}$: Maximum ore storage capacity of j ore pass in a level;

$Q_{ajmin}$: Minimum ore storage capacity of j ore pass in a level;

$Q_{min}$: The minimum tonnage of ore and waste scheduled to be mined during the given shift;

$Q_{max}$: The maximum tonnage of ore and waste scheduled to be mined during the given shift.

### 3.2.3. Variables

$x_{abcij}$: The number of trips of c scraper from i stope to j ore pass in b sublevel, a level;

$y_{abcij}$: The number of trips of c scraper from j ore pass to i stope in b sublevel, a level;

$x_{adj}$: The number of trips of d electric locomotive from j ore pass to the main shaft in a level;

$y_{adj}$: The number of trips of d electric locomotive from the main shaft to j ore pass in a level.

### 3.2.4. Objective

For mine production equipment, long waiting times reduce the production capacity of the mine and increase mining cost. In the production of underground mines, the haulage equipment has the most variable time; therefore, increasing the equipment utilization of the scraper and electric locomotive will indirectly increase the production income of the mine. Equation (9) gives the total waiting time of the scrapers, and Equation (10) gives the total waiting time of the electric locomotives.

$$F_1 = \sum_{a \in \mathcal{A}} \sum_{b \in \mathcal{B}} \sum_{c \in \mathcal{C}_{ab}} \left( T_{limit} - \sum_{i \in \mathcal{N}_{ab}} \sum_{j \in \mathcal{M}_a} z_{abcij} x_{abcij} - \sum_{i \in \mathcal{N}_{ab}} \sum_{j \in \mathcal{M}_a} q_{abcij} y_{abcij} \right) \quad (9)$$

$$F_2 = \sum_{a \in \mathcal{A}} \sum_{d \in \mathcal{D}_a} \left( T_{limit} - \sum_{j \in \mathcal{M}_a} z_{adj} x_{adj} - \sum_{j \in \mathcal{M}_a} q_{adj} y_{adj} \right) \quad (10)$$

When the constraints are met, the model is optimized using the objective function, which can minimize the total waiting time of the scrapers and electric locomotives in the haulage equipment dispatch scheme. The scheme can meet the daily production balance and high-efficiency mining requirements of the mine as well as maximize the utilization rate of the equipment as much as possible.

### 3.2.5. Constraints

Constraints (11) and (12) are the logical limits of the operating time of the scrapers and electric locomotives, respectively. The total operating time of each scraper and electric locomotive during a shift cannot exceed the length of the shift. Constraint (13) ensures that the tonnage of ore hauled from each stope does not exceed the total tonnage of ore in the stope. Constraint (14) ensures that the loading times of any mining site are less than the maximum loading times of the stope within a shift. Constraint (15) ensures that the total ore input in each ore pass does not exceed the maximum tonnage of ore drawn from the ore pass in a shift. Constraint (16) ensures that the ore hauled by the electric locomotive from each ore pass does not exceed the general tonnage of ore drawn from the ore pass in each level. Constraint (17) requires that the tonnage of ore in each ore pass is always within the upper and lower limits of the ore storage of the ore pass. Constraint (18) ensures that the tonnage of ore hoisted by the main shaft does not exceed the total ore output of all

scrapers in a shift. Constraint (19) determines the volume of ore and waste scheduled to be mined during each shift based on the number of shift hours in the plan period determined by the short-term planning period. Constraints (20) and (21) require that the total number of round trips of the scrapers or electric locomotives on each line be equal.

$$\sum_{i \in \mathcal{N}_{ab}} \sum_{j \in \mathcal{M}_a} z_{abcij} x_{abcij} + \sum_{i \in \mathcal{N}_{ab}} \sum_{j \in \mathcal{M}_a} q_{abcij} y_{abcij} \leq T_{limit}, \forall a, b, c \tag{11}$$

$$\sum_{j \in \mathcal{M}_a} z_{adj} x_{adj} + \sum_{j \in \mathcal{M}_a} q_{adj} y_{adj} \leq T_{limit}, \forall a, d \tag{12}$$

$$\sum_{c \in \mathcal{C}_{ab}} \sum_{j \in \mathcal{M}_a} R_{abc} x_{abcij} - g_{abj} \leq 0, \forall a, b, i \tag{13}$$

$$\sum_{c \in \mathcal{C}_{ab}} \sum_{j \in \mathcal{M}_a} x_{abcij} - h_{abj} \leq 0, \forall a, b, i \tag{14}$$

$$\sum_{b \in \mathcal{B}} \sum_{c \in \mathcal{C}_{ab}} \sum_{i \in \mathcal{N}_{ab}} R_{abc} x_{abcij} - J_{aj} \leq 0, \forall a, j \tag{15}$$

$$\sum_{d \in \mathcal{D}_a} R_{ad} x_{adj} - F_{aj} \leq 0, \forall a, j \tag{16}$$

$$Q_{ajmin} \leq \sum_{b \in \mathcal{B}} \sum_{c \in \mathcal{C}_{ab}} \sum_{i \in \mathcal{N}_{ab}} R_{abc} x_{abcij} - \sum_{d \in \mathcal{D}_a} R_{ad} x_{adj} \leq Q_{ajmax}, \forall a, j \tag{17}$$

$$\sum_{a \in \mathcal{A}} \sum_{b \in \mathcal{B}} \sum_{c \in \mathcal{C}_{ab}} \sum_{i \in \mathcal{N}_{ab}} \sum_{j \in \mathcal{M}_a} R_{abc} x_{abcij} \geq \sum_{a \in \mathcal{A}} \sum_{d \in \mathcal{D}_a} \sum_{j \in \mathcal{M}_a} R_{ad} x_{adj} \tag{18}$$

$$Q_{min} \leq \sum_{a \in \mathcal{A}} \sum_{d \in \mathcal{D}_a} \sum_{j \in \mathcal{M}_a} R_{ad} x_{adj} \leq Q_{max} \tag{19}$$

$$\sum_{c \in \mathcal{C}_{ab}} x_{abcij} = \sum_{c \in \mathcal{C}_{ab}} y_{abcij}, \forall a, b, i, j \tag{20}$$

$$\sum_{d \in \mathcal{D}_a} x_{adj} = \sum_{d \in \mathcal{D}_a} y_{adj}, \forall a, j \tag{21}$$

## 4. Optimization Algorithm

Currently, optimization algorithms have become a common method of solving mathematical models, and their development speed is also increasing. According to the variable characteristics of different mathematical models, this paper chooses different optimization algorithms to optimize and solve the corresponding plan. For the short-term resource planning model with binary variables, this article intends to improve the artificial bee colony algorithm for model optimization. The artificial bee colony algorithm has a fast convergence speed. Through the individual's local optimization behavior, the global optimal value will finally emerge in the group. For the haulage equipment dispatch model with multiple continuous integer variables, this paper intends to integrate the non-dominated sorting algorithm with genetic algorithm to optimize the scheduling plan. The chromosome encoding method in genetic algorithm is very suitable for route planning problems. Then, the optimal solution can be found for the multi-objective problem through the non-dominated solution.

### 4.1. Improved Artificial Bee Colony Optimization Algorithm

An artificial bee colony optimization algorithm is a swarm intelligence optimization algorithm inspired by bee colony foraging behavior. This algorithm introduces three types of bees: picking bees, following bees, and scout bees. Different bees perform different tasks in the process of finding an optimal nectar source. The task of picking bees is to

extensively search for nectar sources, perform a neighborhood search for better nectar sources, and determine whether to replace the nectar source according to the comparison of fitness. Following bees select the nectar source after neighborhood search using the roulette method and determine whether to replace the nectar source according to the comparison of fitness. When the nectar source location of the picking bee and the following bee meets the nectar source abandonment condition, they will become the scout bee, and the scout bee will randomly search for a new nectar source at the abandoned nectar source. The specific implementation process of the algorithm is as follows:

(1) Determine the fitness value of the objective function and initialize the parameters, including the nectar population N, the maximum evolutionary generation t, and the custom generation limit;

(2) The coding rules of the nectar source location, the nectar source population adopts binary coding are expressed as $\begin{bmatrix} a_{11} & \cdots & a_{1N} \\ \vdots & \ddots & \vdots \\ a_{m1} & \cdots & a_{mN} \end{bmatrix}$ where m represents the sum of all variable elements of a single individual;

(3) Initialize the nectar population, find a feasible solution according to the constraints of the optimization model, and randomly generate feasible solutions in the surrounding area of the feasible solution. All the generated feasible solutions form the initial nectar population;

(4) Calculate the fitness value of the initial nectar source population, compare the fitness value of the current population, record the best individual value in the current population, and position the honeybees at the half of the nectar source in the population where the fitness value is better. The number of following bees is the same as the number of picking bees;

(5) Picking bees are used to search the neighborhood at the current nectar source location. When the binary code of discrete variables is used, the neighborhood search becomes a value change 0 and 1. After the value is changed, it is judged whether it satisfies the constraint condition. If the constraint condition is not met, the variable is reselected near the value of the variable for transformation until the constraint condition is met, at which point, it can be used as a new nectar location. Then, calculate the fitness value and compare the fitness value of the new nectar source with the original nectar source. If the nectar source quality of the new nectar source is better, replace the original location with the new nectar source to update the nectar source population;

(6) Compare the fitness value of the current population. Compare the current optimal individual with the recorded optimal individual, if it is better than the recorded optimal individual, replace it; otherwise, replace the recorded optimal individual back to the original one. Then, continue subsequent operations at the location;

(7) According to the roulette selection method which is $p_i = fit_i / \sum_{n=1}^{N} fit_n$ following bees choose a better position in the current nectar population and go through the neighborhood search method of step (5) and generate a new nectar source location around this location. Then, the following bee calculates the fitness value, compares the fitness value of the new nectar source with the original nectar source, selects the best to form a new nectar source population, and proceeds to step (6);

(8) Determine whether part of the nectar source in the current nectar source population meets the abandonment condition. If a nectar source has not been replaced after the limit generation neighborhood search, then go to step (9), otherwise go to step (10);

(9) If the nectar source is the best nectar source in the current population, do not abandon it; otherwise, the current nectar source is abandoned, and the picking or following bees at the current nectar source location become the scout bees. The nectar source randomly changes its position to form a new nectar source and updates the nectar source population;

(10) Judge whether the end condition is reached, if the maximum number of iterations is not reached, the current nectar source population is used as the initial nectar source population, and steps (5), (6), (7), (8), and (9) are repeated. If the maximum evolutionary generation is reached, the optimal nectar source position in the current population is considered to be the optimal solution.

*4.2. Non-Dominated Sorting Genetic Algorithm (NSGA-III)*

The elite-level retention strategy is added to the traditional non-dominated sorting genetic algorithm, and the current optimal individual is retained in each iteration. Rapid non-dominated ranking of individuals in the mixed population is used to divide the non-dominated level. The selection of individuals at the same non-dominated level no longer uses crowding degree distance, but uses the constraint dominance relationship based on the reference point to select the elites with the mixed population. In this way, the elite individual in the mixed population is selected to enter the next generation, retaining the superior genes of the parent. The specific description is as follows:

(1) Initialization parameters. The maximum evolutionary generation is $G_{max}$ the reference point size is H, the population size is N + 1, the crossover and mutation rates are $P_c$ and $P_m$, respectively, and the evolutionary generation is t, which is set as 0.

(2) Initialize the $P_t$ population. The scale is N. Each individual in the population consists of N+M chromosomes. The individual chromosomes are independent of each other. The maximum and minimum lengths of chromosomes are set. The length of each chromosome is randomly generated between the maximum and minimum values. The chromosomes are coded using characters, with lowercase for stopes, capital letters for ore passes, and the letter 'Z' for the main shaft. For example: $[aAbBcC]$ represents the operating route of a scraper, while $[AZBZAZ]$ represents the operating route of an electric locomotive.

(3) Determine the objective function. The Pareto ranking hierarchical comparison method is used for multiple objective functions.

(4) Using rapid non-dominated sorting based on Pareto dominance, divide the current $P_t$ population into several dominance layers, select the best individuals in the first dominance layer based on the reference point-based constraint dominance relationship method, and extract them as a single population that does not participate in genetic manipulation. Then, subpopulation $B_t$ is generated through the crossover, mutation, and breaking of the genetic algorithm, with a scale of N. Crossover: randomly select chromosomes of the same nature for different individuals in the population to cross over random gene positions. Mutation: Chromosomal genetic properties mutate at random. Breaking: Each chromosome randomly chooses whether to perform the break operation, if so, cut two genes to the last position of the chromosome. The progeny population is generated through the operation sequence of crossover, mutation, and breaking.

(5) Combine the $B_t$ offspring population and the $P_t$ parent population to form the $R_t$ population, use the non-dominated sort based on Pareto dominance to divide $R_t$ into several different non-dominated layers, and select N higher-level individuals as the next $P_{t+1}$ parent population. Individuals in the same level are selected using a reference point-based constraint dominance relationship method, also the best individual in the first dominance level of the $P_{t+1}$ parent population is selected using the same method.

(6) Compare and select the optimal individual produced by the t + 1 generation with the optimal individual of the t generation. Use the method based on the reference point to select the superior individual among the two adjacent generations of optimal individuals. If the optimal individual of the t + 1 generation is superior, it will be placed in a separate population, and the original t generation optimal individual will be placed in t, where the best individual of the t + 1 generation is located, and form a new $P_{t+1}$ parent group.

(7) Judge whether the $P_{t+1}$ parent population meets the termination conditions. If not, then t = t + 1, repeat steps (4), (5), and (6). If the termination conditions are met, then output the best individual.

## 5. Computational Study

In the following sections, the optimization scheme is compared and verified with the scheme obtained using the traditional optimization algorithm. The mathematical model, as well as the algorithm's high efficiency and feasibility, is described.

### 5.1. Optimization Results

First, the actual production data of the mine are brought into the bee colony algorithm to solve the short-term resource plan. Then, according to the spatial location of the stopes and ore passes to be mined in the short-term resource plan, the time parameters in the haulage equipment dispatch plan model are obtained, when are subsequently brought into non-dominated sorting genetic algorithm to calculate the haulage equipment scheduling scheme.

#### 5.1.1. Resource Plan

The calculation results of the resource planning model are shown in Table 1. The number '1' indicates that the site can be mined during the plan period; on the contrary, the number '0' indicates that the site cannot be mined during the plan period. The number of the ore blocks in Table 1 are shown in Figure 5. For example, (1, 1, 4, 3) means the ore block numbered (4, 3) in the plan view in (1, 1) sublevel.

**Table 1.** Calculation results.

| Number | Result | Total Storage/t | Remaining Storage/t | Grade/% |
|---|---|---|---|---|
| (1, 1, 4, 3) | 1 | 2883.92 | 2883.92 | 24.03 |
| (1, 1, 8, 3) | 1 | 2951.56 | 2951.56 | 61.74 |
| (1, 1, 11, 3) | 1 | 2631.26 | 2631.26 | 44.26 |
| (1, 1, 1, 2) | 1 | 2951.37 | 2951.37 | 36.46 |
| (1, 1, 6, 2) | 1 | 2601.16 | 2601.16 | 49.15 |
| (1, 1, 9, 1) | 1 | 2667.56 | 2667.56 | 49.28 |
| (1, 1, 12, 1) | 1 | 2916.49 | 2916.49 | 56.74 |
| (2, 1, 1, 3) | 1 | 2643.29 | 2643.29 | 35.77 |
| (2, 1, 4, 3) | 1 | 2708.67 | 2708.67 | 42.90 |
| (2, 1, 10, 3) | 1 | 2947.29 | 2947.29 | 31.48 |
| (2, 1, 2, 1) | 1 | 2938.19 | 2938.19 | 52.25 |
| (2, 1, 5, 1) | 1 | 2646.24 | 2646.24 | 54.64 |
| (2, 1, 8, 1) | 1 | 2951.56 | 2951.56 | 61.74 |
| (2, 1, 11, 1) | 1 | 2969.06 | 2969.06 | 44.26 |
| (2, 1, 7, 3) | 1 | 2671.67 | 2671.67 | 48.00 |
| Others | 0 | / | / | / |

#### 5.1.2. Haulage Equipment Dispatch Plan

The short-term resource plan indicates that there are 15 stopes in the plan period, seven stopes in (1, 1) sublevel, and eight stopes in (2, 1) sublevel. According to the production arrangement of the mine, there are two ore passes in the first level and three ore passes in the second level during the plan period. The travel time parameters of the scrapers and electric locomotives measured by the mine are shown in Tables 2–7.

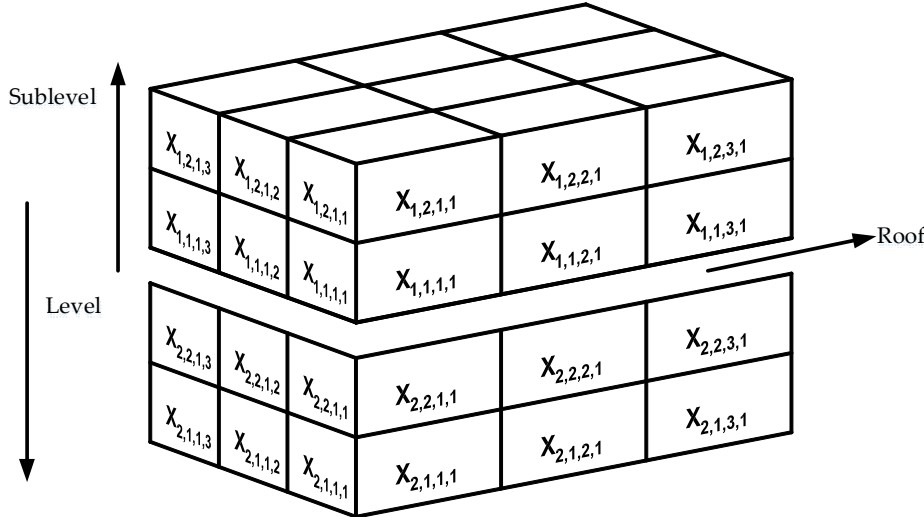

**Figure 5.** Simple spatial model of ore blocks.

The total income during the plan period is 5,325,966 dollars.

**Table 2.** Loaded travel time of the scraper between the stope and ore pass in (*1, 1*) sublevel/s (Units: s).

| Ore Pass | Stope/(Number) | | | | | | |
|---|---|---|---|---|---|---|---|
| | **a** | **b** | **c** | **d** | **e** | **f** | **g** |
| | **(1, 1, 4, 3)** | **(1, 1, 8, 3)** | **(1, 1, 11, 3)** | **(1, 1, 1, 2)** | **(1, 1, 6, 2)** | **(1, 1, 9, 1)** | **(1, 1, 12, 1)** |
| A | 75.5 | 149.3 | 179.2 | 97.8 | 100.5 | 145.6 | 210.3 |
| B | 154.2 | 85.6 | 96.2 | 205.2 | 137.3 | 79.7 | 100.5 |

**Table 3.** Empty travel time of the scraper between the stope and ore pass in (*1, 1*) sublevel/s (Units: s).

| Ore Pass | Stope/(Number) | | | | | | |
|---|---|---|---|---|---|---|---|
| | **a** | **b** | **c** | **d** | **e** | **f** | **g** |
| | **(1, 1, 4, 3)** | **(1, 1, 8, 3)** | **(1, 1, 11, 3)** | **(1, 1, 1, 2)** | **(1, 1, 6, 2)** | **(1, 1, 9, 1)** | **(1, 1, 12, 1)** |
| A | 52.49 | 105.7 | 147.5 | 64.5 | 67.8 | 104.3 | 172.6 |
| B | 110.7 | 57.5 | 61.3 | 144.2 | 84.6 | 54.0 | 68.2 |

**Table 4.** Loaded travel time of the scraper between the stope and ore pass in (*2, 1*) sublevel/s (Units: s).

| Ore Pass | Stope/(Number) | | | | | | |
|---|---|---|---|---|---|---|---|
| | **s** | **t** | **u** | **v** | **w** | **x** | **y** | **z** |
| | **(2, 1, 1, 3)** | **(2, 1, 4, 3)** | **(2, 1, 7, 3)** | **(2, 1, 10, 3)** | **(2, 1, 2, 1)** | **(2, 1, 5, 1)** | **(2, 1, 8, 1)** | **(2, 1, 11, 1)** |
| C | 87.3 | 86.7 | 144.7 | 173.7 | 91.8 | 114.7 | 152.9 | 177.4 |
| D | 157.6 | 133.3 | 92.3 | 132.7 | 145.7 | 87.2 | 67.1 | 134.7 |
| E | 214.7 | 195.7 | 113.8 | 56.4 | 215.3 | 187.3 | 105.4 | 80.5 |

**Table 5.** Empty travel time of the scraper between the stope and ore pass in (*2, 1*) sublevel/s (Units: s).

| Ore Pass | Stope/(Number) | | | | | | |
|---|---|---|---|---|---|---|---|
| | **s** | **t** | **u** | **v** | **w** | **x** | **y** | **z** |
| | **(2, 1, 1, 3)** | **(2, 1, 4, 3)** | **(2, 1, 7, 3)** | **(2, 1, 10, 3)** | **(2, 1, 2, 1)** | **(2, 1, 5, 1)** | **(2, 1, 8, 1)** | **(2, 1, 11, 1)** |
| C | 54.5 | 54.2 | 106.7 | 129.8 | 56.3 | 73.2 | 111.4 | 132.3 |
| D | 110.3 | 89.6 | 56.7 | 90.5 | 107.3 | 55.1 | 43.8 | 92.3 |
| E | 162.3 | 141.4 | 72.7 | 34.3 | 160.9 | 137.9 | 69.6 | 51.7 |

**Table 6.** Loaded travel time of the electric locomotive between ore pass and the main shaft/s (Units: s).

| Main Shaft | Ore Pass | | | | |
|---|---|---|---|---|---|
| | A | B | C | D | E |
| Z | 900 | 963 | 935 | 973 | 892 |

**Table 7.** Empty travel time of the electric locomotive between the ore pass and main shaft/s (Units: s).

| Main Shaft | Ore Pass | | | | |
|---|---|---|---|---|---|
| | A | B | C | D | E |
| Z | 725 | 772 | 741 | 694 | 711 |

The short-term resource planning scheme and running time parameters of the above scrapers and electric locomotives are brought into MATLAB to create the haulage equipment scheduling plan. The change curve of the target optimization value with the evolutionary generation is shown in Figure 6, where Object 1 represents F1, Object 2 represents F2, the x-axis is generations, and the y-axis is fitness value. The calculation results of variables in the final haulage equipment dispatch plan model are shown in Tables 8 and 9.

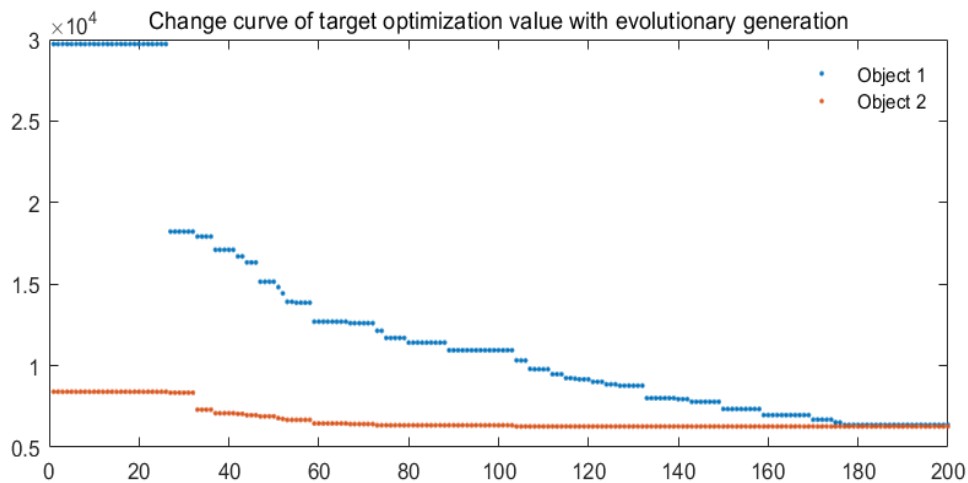

**Figure 6.** Change curve of the target optimization value with evolutionary generation.

**Table 8.** Loaded haulage number of times of the scraper (Units: 1).

| Ore Pass | Stope | | | | | | | | | | | | | | |
|---|---|---|---|---|---|---|---|---|---|---|---|---|---|---|---|
| | a | b | c | d | e | f | g | s | t | u | v | w | x | y | z |
| A | 19 | 20 | 26 | 20 | 26 | 30 | 35 | / | / | / | / | / | / | / | / |
| B | 21 | 13 | 16 | 28 | 15 | 18 | 15 | / | / | / | / | / | / | / | / |
| C | / | / | / | / | / | / | / | 9 | 14 | 7 | 17 | 9 | 16 | 19 | 17 |
| D | / | / | / | / | / | / | / | 13 | 9 | 6 | 13 | 8 | 12 | 6 | 7 |
| E | / | / | / | / | / | / | / | 14 | 4 | 10 | 6 | 15 | 13 | 7 | 6 |

**Table 9.** Unloaded haulage number of times of electric locomotives (Units: 1).

| Main Shaft | Ore Pass | | | | |
|---|---|---|---|---|---|
| | A | B | C | D | E |
| Z | 6 | 7 | 6 | 5 | 5 |

In order to prove the superiority of the non-dominated sorting genetic algorithm, the calculation results of the non-dominated sorting genetic algorithm are compared with the results of the traditional genetic algorithm. Solving a multi-objective problem using the traditional genetic algorithm method involves transforming the multi-objective problem into a single objective by weighting to select the superior individual and generating the superior scheme. Since the single running time of the electric locomotive is longer than that of the scraper, the proportion of F2 is relatively low, at 10%, while the proportion of F1 is 90%. The change curve of the target value in the calculation process with the evolutionary generation is shown in Figure 7. The results of comparison between traditional scheme and the scheme in Tables 8 and 9 are shown in Table 10.

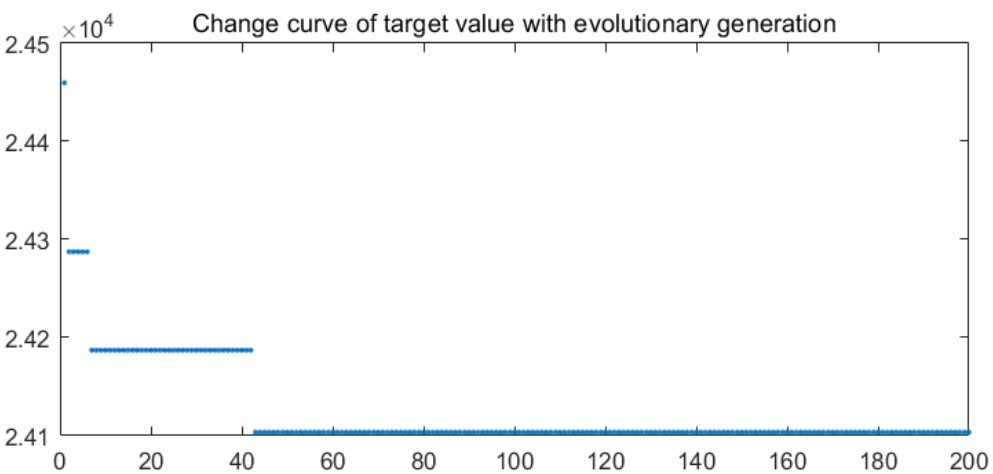

**Figure 7.** Change curve of the target value with evolutionary generation.

**Table 10.** Comparison of the two schemes.

| Method | The Tonnage of Ore Transported by the Scraper/t | The Tonnage of Ore Transported by the Electric Locomotive/t | Total Waiting Time/h |
|---|---|---|---|
| NSGA-III | 1677 | 1508 | 3.5 |
| GA | 1118 | 1005 | 11 |

*5.2. Result Analysis*

The change curve in Figure 5 reflects the change trend of the multi-target value of the optimal individual in the evolution process. The multi-target value of the optimal individual in each generation changes significantly, and when it evolves to approximately 180 generations, the curve tends to stable, at which point the optimal solution of the multi-objective optimization problem is obtained. The optimal target value curve of the traditional genetic algorithm in Figure 6 is relatively simple and hierarchical. The curve tends to be stable when it evolves to approximately 40 generations, indicating that the optimization intensity is insufficient in the calculation process, and the optimization result is relatively rough.

In the scheme calculated by the NSGA-III-algorithm, the tonnage of ore hauled by the scraper from the stope is 1677 t, and the tonnage of ore hauled by the electric locomotive to the main shaft is 1508 t during a shift (Table 10). The wait time is 3.5 h. In the scheme calculated using the traditional genetic algorithm, the total wait time is 11 h, the tonnage of ore hauled from the stope is 1118 t, and the tonnage of ore hauled to the main shaft is 1005 t. In the better scheme, the total wait time of the scraper and electric locomotive was reduced by 7.5 h, the ore hauled from the stope increased by 559 t, and the ore hauled to the main shaft for hoisting increased by 503 t. The non-dominated sorting genetic algorithm has better performance than the traditional genetic algorithm and has higher practical value.

The operation route of the scraper is not limited to a single stope and ore pass (Figure 8). The simultaneous digging and loading operation of multiple stopes can improve the smoothness of the succession of each process. Through this operation mode, the U-turn and wait time of the scraper at the ore pass are reduced.

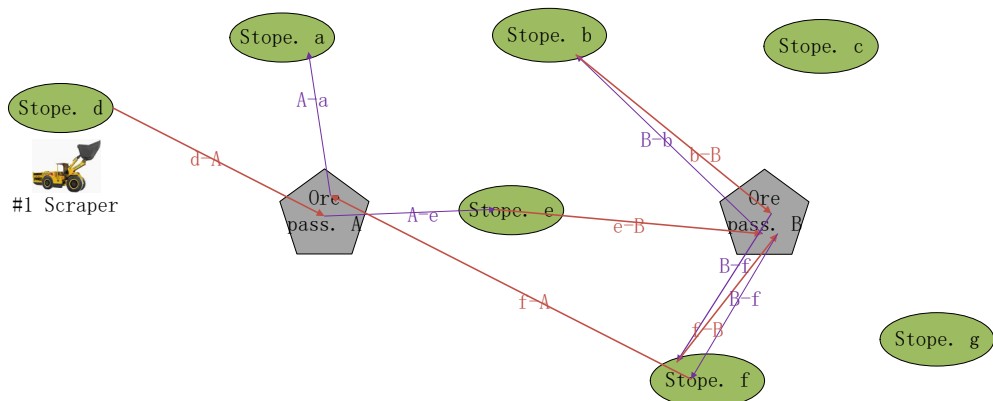

**Figure 8.** Operation route of the #1 scraper during a certain period of time.

For the studied mine, the constraints considered are insufficient when formulating a short-term resource plan. The constructed resource plan barely meets the mine's production needs and fails to maximize the mine's production benefits. The haulage equipment dispatch plan prepared on this basis is inefficient. In this paper, the equipment dispatch plan is based on the short-term resource plan. From the optimization results, the short-term resource plan is used to guide the dispatching plan of the scraper and the electric locomotive, which can efficiently complete the short-term production tasks. Therefore, as far as the traditional scheduling scheme of the mine is concerned, this research can provide a comprehensive and excellent production decision-making scheme based on meeting the production requirements of the mine, which can improve the utilization rate of equipment, and is suitable for production and haulage in the mine under study.

## 6. Discussion

Through the mine short-term resource planning optimization model, the operating location of the scraper is obtained. Then, the dispatching plan of the scraper is prepared using the dispatching model. This article only arranges the dispatching plan of the scraper in a certain period of time, and does not realize the preparation of the dispatching plan of the scraper for the entire short-term planning cycle. Therefore, the follow-up research can be conducted to further improve the entire short-term dynamic dispatching plan of the scraper. In addition, the size of the population limits its global search, so the population size can be increased appropriately, and the parameters in the algorithm flow can also be appropriately optimized.

## 7. Conclusions

Based on 0-1 integer programming, combined with the production needs of underground mines under study, a short-term resource plan optimization model for underground mines is constructed with the maximum profit as the primary objective. The constraints of this model conform to the actual conditions of the mine. The artificial bee colony optimization algorithm is used to solve the model, and the requirements of the artificial bee colony for the nectar source are restricted by the constraints. The neighborhood search method can be used to find the optimal nectar source over a wide range. The addition of the following bee and scout bee can improve the optimization results. Then, based on the short-term resource plan, considering the constraints of each link in the underground mine ore haulage process, the shortest total equipment wait time is used as the objective function to construct an optimization model for haulage equipment between stopes, which

can more completely describe the process of dynamic allocation of the scraper and electric locomotives. Using a non-dominated sorting genetic algorithm, a more accurate Pareto optimal solution set can be obtained by introducing elite selection strategy and selection methods based on reference points. Finally, verified by examples, this paper can provide a basis for the formulation of a short-term plan and equipment dispatch decisions for underground mines, and improve the mine's production revenue and equipment utilization. Future research directions should focus on the preparation of dispatching planning for multiple pieces of equipment and should not be limited to haulage equipment.

**Author Contributions:** Conceptualization, N.L.; Data curation, H.Y.; Formal analysis, S.F. and D.C.; Funding acquisition, N.L., Q.W. and M.J.; Investigation, Q.W.; Methodology, N.L. and M.J.; Resources, H.Y., S.Z. and L.W.; Supervision, N.L.; Validation, S.F. and S.Z.; Writing—original draft, S.F.; Writing—review & editing, H.Y., Q.W., S.Z., M.J., L.W. and D.C. All authors have read and agreed to the published version of the manuscript.

**Funding:** This research is supported by the National Key R&D Program of China (Grant No. 2019YFC0605304), National Natural Science Foundation of China (Grant No. 51704218), and Special Fund Project of the Basic Scientific Research Operation Fee of the Central University (Grant No. 2019III086CG and 2018IVB054).

**Data Availability Statement:** Not applicable.

**Acknowledgments:** Thanks to the relevant departments for funding and the mining companies that provided the experiment.

**Conflicts of Interest:** The authors declare no conflict of interest.

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
