# Peer review of "Dispatch Optimization Model for Haulage Equipment between Stopes Based on Mine Short-Term Resource Planning"

_metals, doi:10.3390/met11111848_

Round 1

Reviewer 1 Report

Please refer to the attached annotated pdf copy of the paper for your reference. Parts of the paper have incorrect technical mining terminology making it a challenge to follow the technical reasoning presented in the paper. Figure 2 is meaningless. Some statements are made without any convincing technical justification. The term 'volume' is incorrectly used in Section 3.1.2 to imply 'tonnage'.  Constraints (3) and (4) are problematic and should be checked for correctness. The findings are superficial and lack practical application. Equipment allocation and dispatching are two intricately related optimization decisions. If equipment allocation is incorrectly done then the equipment dispatch inherently carries over errors made in the allocation stage, resulting in sub-optimal equipment dispatching decisions and this view is not clearly articulated in the paper.

Author Response

Point 1: In section 3.1.2, “Total volume of ore and waste”.

Response 1: Change "volume" to "tonnage". And I think there is no density parameter required.

Point 2: In section 3.1.2, “ : Mineral processing recovery ratio of the metal”.

Response 2: I mean that dressing recovery ratio of the metal of dressing plant. It’s different from ore recovery ratio.

Point 3: In section 3.1.2, “Total volume of ore and waste”.

Response 3: The total number of the two types of equipment is used to ensure that the equipment does not interfere with each other, and to reduce the waiting time of each equipment.

Point 4: In section 3.1.5

Response 4: Add the “R” is because that there will be losses when the ore is mined in each stope.

Point 5: In section 3.1.5

Response 5: The left side of the formula is summed and compared with the right side.

Point 6: In section 3.2.2

Response 6: The unit of “capacity” is “t”.

Point 7: In section 3.2.5, “Constraint (18)”.

Response 7: I see that, but I cannot guarantee that this limit will not be exceeded when calculating.

Reviewer 2 Report

Title: Dispatch optimization model for haulage equipment…

Manuscript ID: metals-1449335

Authors: Li et al.

Dear Authors,

Thank you for the opportunity to read your article. I found the topic is interesting and potentially useful. Generally speaking, there are some results presented in order to capture some trends. On the other hand, some restructurings of introduction and literature review as well as method section are required. The results need more deeper discussion and clear explanation. I suggest that this article will be revised before resubmission for another review process. As a conclusion, I recommend its major revision at this state.

I hope my comments are helpful.

Good luck,

A reviewer

Major concerns:

“Keywords”

-Please consider listing keywords that are not used in the article title.

“1. Introduction and 2. Relevant Literature”

-Please provide the research gaps that you tried to address in this work. As you identified there are a lot of previous works reported, but there are no specific research gaps mentioned in the introduction.

-Please clearly mention the objectives of your study.

-The two sections consume a large volume of the paper, i.e. 7 out of 21 pages. Please try to limit their contents and mention the contents directly relevant to your study.

“3. Model”->3. Models

“3.1.2. Parameters”->Please consider providing the definitions of the parameters.

-Did you use actual values from experiments/literature? Or did you generate the values of those parameters by yourself? Please state this point in this section with necessary literature.

-You have only one P for the Market price of metal. In your model, did you consider only one type of metal contained in an ore? If yes, please state so in this section.

“4. Computational study”

“4.1 Optimization algorithm”->Please consider sending this section to method section since it explains some method but does not contain any results.

-Lines 662-663: “The simultaneous digging and loading operation of multiple slopes can improve the smoothness of the succession of each process.”->(a) This sounds reasonable, but what are the other options that do not maximize the operation? It would be useful to compare different options and their performance. (b) Maybe you have already tried to do so as shown in Tables 1 to 7, but the explanations of operation conditions are not clear at least to me. For example, what are the numbers in brackets in those tables (e.g. (1,1,4,3))?

-Fig.7: Please make letters larger. The current figure is very difficult to read.

Author Response

Point 1: In section 1, “Previous scholars' research on short-term production in underground mines only divided the mining areas of the mine at each period, and did not involve the actual mining process.  The production dispatching planning of underground mining vehicles is a short-term production planning process that is different from short-term re-source planning. Only after short-term resource planning is completed, can production dispatching planning begin. However, most short-term production planning does not involve precise equipment dispatch path optimization research. This article combines resource planning and dispatching path optimization to guide the actual short-term production of a mine.”

Response 1: In this paragraph, I mentioned some research gaps and the research purpose of this article.

Point 2: In section 2.

Response 2: Some content has been cut down, and I look forward to further comments from expert reviewer.

Point 3: “3.1.2 Parameters”->Please consider providing the definitions of the parameters.

-Did you use actual values from experiments/literature? Or did you generate the values of those parameters by yourself? Please state this point in this section with necessary literature.

Response 3: The definition of some parameters has been corrected. In addition, the parameter values in the subsequent calculation process depend on the parameter data of the equipment, stope and ore pass in the actual mining process.

Point 4: You have only one P for the Market price of metal. In your model, did you consider only one type of metal contained in an ore? If yes, please state so in this section.

Response 4: This article only uses the market price of iron.

Point 5: -Lines 662-663. “The simultaneous digging and loading operation of multiple stopes can improve the smoothness of the succession of each process.”-> (a) This sounds reasonable, but what are the other options that do not maximize the operation? It would be useful to compare different options and their performance.

Response 5: Simultaneous operation of multiple stopes can ensure the balance of production and increase the utilization rate of equipment. However, the previous research rarely involves the equipment scheduling problem of underground mines. Therefore, the advantages of open-pit mine equipment scheduling are applied to underground mines.

Point 6: (b) Maybe you have already tried to do so as shown in Tables 1 to 7, but the explanations of operation conditions are not clear at least to me. For example, what are the numbers in brackets in those tables (e.g. (1, 1, 4, 3)).

Response 6: The Number of the ore blocks in Table 1 are shown in Figure 5. For example (1, 1, 4, 3), it means the ore block numbered (4, 3) in the plan view in (1, 1) sublevel.

Point 7: The results need more deeper discussion and clear explanation.

Response 7: 6 Discussion

Through the mine short-term resource planning optimization model, the operat-ing location of the scraper is obtained. Then the dispatching plan of the scraper is pre-pared using the dispatching model. This article only arranges the dispatching plan of the scraper in a certain period of time, and does not realize the preparation of the dis-patching plan of the scraper for the entire short-term planning cycle. Therefore, the follow-up research can be further improved. The entire short-term dynamic dispatch-ing plan of the scraper can be prepared more excellently. In addition, the size of the population limits its global search, so the population size can be increased appropri-ately, and the parameters in the algorithm flow can also be appropriately optimized.

Reviewer 3 Report

It is a very complete theoretical paper

It must be corrected:

Key words are very long, they must be simplified

Line 95: Indicate on a map of China the location of the aforementioned mine and its surroundings, since its facilities are mentioned in the calculation.

Figures 1 and 2 are not cited in the text.

Line 357: Constraint eq. (2)? The same lines 359, 363, 366, 369, 373, 375, 448 and following. 

Author Response

Point 1: Key words are very long, they must be simplified.

Response 1: “Keywords: digital mine; mine short-term production planning; haulage equipment dispatch plan; ABCA; NSGA”

Point 2: Line 95: Indicate on a map of China the location of the aforementioned mine and its surroundings, since its facilities are mentioned in the calculation.

Response 2: I have added the location map of the iron mine as Figure 1.

Point 3: Figure 1 and 2 are not cited in the text.

Response 3: I have corrected the original Figure 1 and Figure 2.

Point 4: Line 357: Constraint eq. (2)? The same lines 359, 363, 366, 369, 373, 375, 448 and following.

Response 4: I have corrected some expression of constraints, and I look forward to further comments from expert reviewer.

Round 2

Reviewer 2 Report

Dear Authors,

Thank you for your effort to address the comments and concerns.

I see they were all addressed, and suggest that the journal accepts the manuscript for its publication.

Best regards,

A reviewer

Author Response

Dear Reviewer,

Thank you for your comments concerning our manuscript.

Best regards,

All Authors

Reviewer 3 Report

After the corrections, the work has improved a lot. 

Is OK

Author Response

(The authors gave the same response as above.)
